# Experimental Crossing Confirms Reproductive Isolation between Cryptic Species within *Eulimnogammarus verrucosus* (Crustacea: Amphipoda) from Lake Baikal

**DOI:** 10.3390/ijms231810858

**Published:** 2022-09-17

**Authors:** Polina Drozdova, Alexandra Saranchina, Ekaterina Madyarova, Anton Gurkov, Maxim Timofeyev

**Affiliations:** 1Institute of Biology, Irkutsk State University, 664025 Irkutsk, Russia; 2Baikal Research Centre, 664011 Irkutsk, Russia

**Keywords:** cryptic variation, reproductive isolation, speciation, ancient lakes, Baikal, Crustacea, species flocks

## Abstract

Ancient lakes are known speciation hotspots. One of the most speciose groups in the ancient Lake Baikal are gammaroid amphipods (Crustacea: Amphipoda: Gammaroidea). There are over 350 morphological species and subspecies of amphipods in Baikal, but the extent of cryptic variation is still unclear. One of the most common species in the littoral zone of the lake, *Eulimnogammarus verrucosus* (Gerstfeldt, 1858), was recently found to comprise at least three (pseudo)cryptic species based on molecular data. Here, we further explored these species by analyzing their mitogenome-based phylogeny, genome sizes with flow cytometry, and their reproductive compatibility. We found divergent times of millions of years and different genome sizes in the three species (6.1, 6.9 and 8 pg), further confirming their genetic separation. Experimental crossing of the western and southern species, which are morphologically indistinguishable and have adjacent ranges, showed their separation with a post-zygotic reproductive barrier, as hybrid embryos stopped developing roughly at the onset of gastrulation. Thus, the previously applied barcoding approach effectively indicated the separate biological species within *E. verrucosus*. These results provide new data for investigating genome evolution and highlight the need for precise tracking of the sample origin in any studies in this morphospecies.

## 1. Introduction

Unlike seas, most lakes are relatively short-lived, as they tend to fill up with sediments and cease to exist. The absolute majority of the extant lakes emerged within the Holocene epoch (i.e., no earlier than 18,000 years ago) [1]. However, there are some ancient lakes that are much older. The thresholds for considering a lake ancient differ, with a common one being that a lake needs to exist for at least one glacial cycle, i.e., to be at least 130,000 years old [2]. Most of such lakes are located in rift valleys, where tectonic activity counteracts sediment formation [1]. The ancient lakes host fascinating endemic species flocks, or assemblages, which are groups of multiple closely related species [3]. This is also true about Lake Baikal, which is one of the oldest lakes on Earth, tracing back 25–30 million years [4] or even 70 million years, to the Late Cretaceous epoch, when multiple lakes that would later form Baikal emerged [5]. It has a rich invertebrate fauna of >2500 species, most of which are endemic to the lake [6]. One of the most species-rich groups in the lake are gammaroid amphipods (Crustacea: Amphipoda: Gammaroidea), which reach the diversity of over 350 morphological species and subspecies. They are distributed into up to 7 or 10 families according to different authors [7], and, according to the molecular data, are all nested within the genus *Gammarus* [8,9].

When speaking about biodiversity, it is crucial to set up the terminology framework to discuss aspects of defining a species. Here, we will only speak about the object of this work and related organisms, which are strictly sexually reproducing animals.
According to the biological species concept, we will define the *biological species* (or simply species) as the multitude of actually or potentially interbreeding populations, which are reproductively isolated from other such groups [9,10]. Thus, the main criterion for defining a biological species is the presence of a reproductive barrier. The mechanisms of reproductive isolation fall into two groups, prezygotic (mate discrimination, timing of mating and gamete release, fertilization barriers, *etc.*) and postzygotic (hybrid inviability or sterility) [11,12]. As a species may comprise potentially (not actually) interbreeding populations by definition, then geographic isolation does not qualify as a reproductive barrier [10].Then, we will use the term *morphological species*, or *morphospecies*, for the set of individuals with indistinguishable morphological traits. Usually, it is impossible to say if there are any reproductive barriers between such individuals if they are sampled in different locations.Finally, we suggest using the term *barcoding species* for taxonomic entities separated on the basis of at least one phylogenetic marker sequence with at least one species delimitation method. This is mostly equivalent to the term *molecular operational taxonomic unit* (MOTU) or *genospecies* [13]. If there are particular sequence-based (or allozyme-based) clusters but species delimitation did not separate them, was not applied or was not applicable, we will call them *barcoding lineages* or *allozyme lineages*, respectively. Sequence-based delimitation indicates that there is some degree of separation, but may not necessarily mean genetic incompatibility.If one morphological species accommodates several barcoding species, we will term these genetically diverse but morphologically indistinguishable entities *cryptic species*, as suggested in [9]. Similarly, if several barcoding lineages are contained within a morphological species, it is logical to call them *cryptic lineages*. If a morphological difference is found after closer examination, the lineages or species become *pseudocryptic* (e.g., [14]). It is worth noting that cryptic lineages or species may occur both sympatrically and allopatrically, and they may originate either in the process of diversification or by convergent evolution of close (but not necessarily sister) groups [9,15].

The importance of cryptic species is that, in particular, the widely distributed morphological species may in fact be comprised of a multitude of local endemics, which may differ in their resistance against adverse environmental factors [9,16]. Moreover, such difference in pollutant sensitivity was indeed demonstrated, for example, for sympatric cryptic species of copepods [17]. Another study found differential sensitivity to a fungicide and an insecticide between the cryptic species within the amphipod morphospecies *G. fossarum* Koch, 1836, even though it was hard to unambiguously attribute the difference to genetic difference or life history because different populations were connected in different locations [18]. Moreover, a difference in susceptibility to acathocephalan parasites was shown between sympatric cryptic species of the *G. fossarum/ G. pulex* complex [19].

Recent years have seen a significant accumulation of the data on the existence of cryptic species within many taxa, including gammaroid amphipods [20,21,22,23,24,25,26,27]. For example, the number of barcoding species within the *G. fossarum* species complex increased within the last two decades from three in Central Europe [28] to 32–152 (depending on the delimitation method) in the continental-wide assessment [29]. For *G. lacustris* Sars, 1863, a species widely distributed throughout the Holarctic, an impressive number of 119 barcoding species worldwide was described by the sequences of seven marker genes [30]. However, it is important to keep in mind that the barcoding species delimitation based solely on cytochrome oxidase subunit I (COI) may drastically overestimate the number of species [31].

The real extent of cryptic diversity within Baikal amphipods is not yet understood, even though the evidence of cryptic species or lineages has been accumulating. For instance, allozyme analysis uncovered cryptic lineages between presumably conspecific *Pallasea* Spence Bate, 1862 individuals [32]. Similarly, local populations of a widely distributed shallow-water species *E. cyaneus* (Dybowsky, 1874) were revealed to be cryptic allozyme lineages [33]. In yet another species distributed in the littoral throughout the lake, *Gmelinoides fasciatus* (Stebbing, 1899), four cryptic barcoding lineages have been found with cytochrome oxidase subunit I (COI) sequence analysis [34,35]. A phylogeography-related distribution of pseudocryptic lineages was demonstrated for *Brandtia (Dorogostajskia) parasitica* (Dybowsky, 1874), a highly specialized epibiont of sponges in Baikal [36]. Several cryptic lineages with phylogeographic subdivision were found within two species of the genus *Acanthogammarus* Stebbing, 1899, which encompasses nectobenthic amphipods dwelling up to 250-m depths [37]. We found cryptic species within two more *Eulimnogammarus* morphospecies, *E. verrucosus* and *E. vittatus*; in the case of the former, two phylogenetic markers, COI and 18S ribosomal RNA fragments, supported the differentiation [38]. Finally, there is sporadic transcriptome-level finding: three pairs of morphological conspecifics sampled in different places had varying degrees of genetic differences according to their whole transcriptomes [39].

Apart from allozymes, phylogenetic marker sequencing, and high-throughput sequencing techniques, there is one more measurable genome parameter possibly indicative of reproductive isolation, which is genome size. In amphipods, the genome size has been shown to correlate with latitude [40] and body size [41,42]. In addition, two-fold differences in genome size within the *Hyalella azteca* (Saussure, 1858) species complex was found, which correlated with both the ecomorph (large-bodied individuals had larger genomes) and the phylogenetic structure of the four cryptic species [43]. In the highly diverse morphospecies *G. lacustris* (see above), genome sizes of 14.06 pg (boreal lake in Canada) and 8.50 pg (different water bodies throughout North America) were reported by different authors [43,44]. In Baikal amphipods, correlation of genome size with both body size and habitat depth was shown [45], but intraspecies variation has not been assessed.

In this work, we focused on *E. verrucosus* (Gerstfeldt, 1858), which is a relatively large (up to 36 mm adult length [46]) and highly abundant species, shaping littoral macrozoobenthic communities in the lake [47]. It is also found in the Angara river (the only outflow of the lake) [48,49,50] and has even been found in the Yenisei river, into which Angara flows [51]. Recently, we found that *E. verrucosus* includes at least three cryptic species in Lake Baikal, named the western (W), southern (S), and eastern (E), differing in COI and 18S rRNA sequences. A closer re-examination led to the discovery of two morphological features discriminating the eastern species from the other two: (1) the numbers of setae on the dorsal parts of the metasome and the urosome was evidently lower and (2) intermittent black stripes along the caudal edges of the body segments of the individuals (they were continuous in the specimens from the other two species (Figure 1A) [38]. Thus, the western and southern species should be considered cryptic, and the eastern species should be formally considered pseudocryptic. As these species have non-overlapping ranges, it was not evident whether there are any reproductive barriers between them.

Most of the works using *E. verrucosus* focus on the western barcoding species [52,53,54,55,56,57,58,59,60]. For this species, whole genome sequencing was performed, and the genome size estimate based on the coverage of selected single-copy genes equaled 9.96 Gb [61] (10.18 pg if converted to mass units [62]). The cytology-based assessment was 6.10 pg (with Feulgen image densitometry) or 7.13 pg (with flow cytometry) [45], but the sampling location of these specimens was not mentioned. The observed discrepancy could have at least two explanations. First, one of the methods may have given an erroneous estimate. Second, it is conceivable that the reason was that different cryptic species were analyzed.

**Figure 1 ijms-23-10858-f001:**
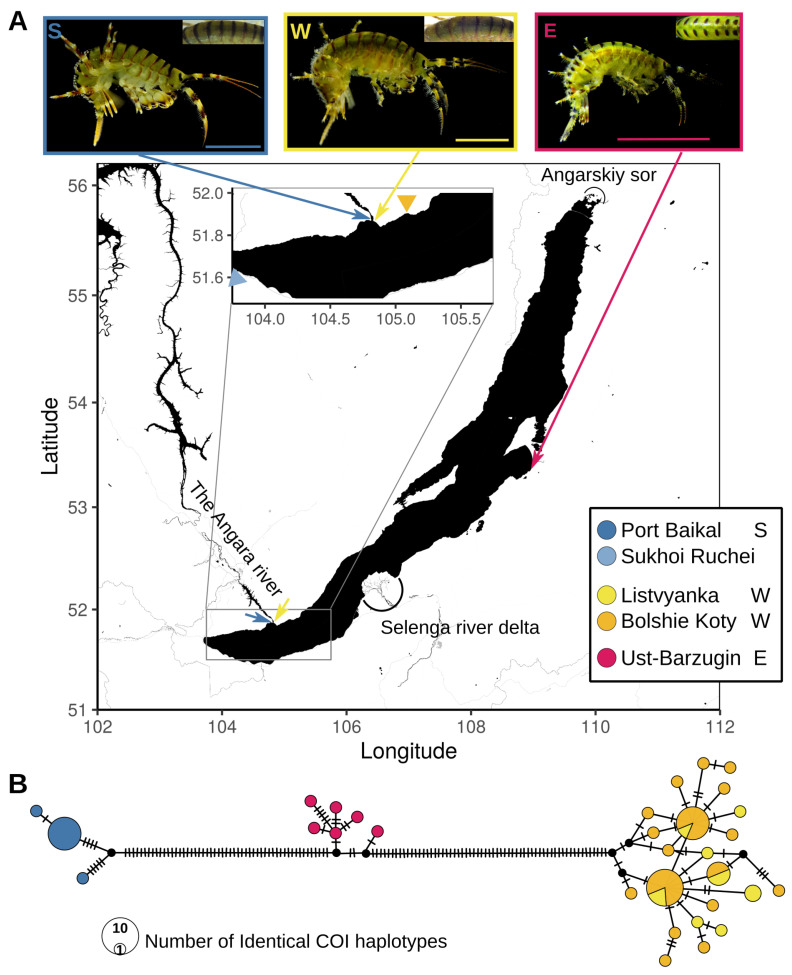
Typical morphology, sampling locations, and molecular diversity of the three (pseudo)cryptic species within *E. verrucosus*. (**A**) Typical morphology of the three species and sampling locations of the specimens used in this work. Samples for flow cytometry and crossing experiments were collected at the locations indicated with colored arrows (blue, Port Baikal (S); yellow, Listvyanka (W); magenta, Ust-Barguzin (E)). An additional orange triangle arrowhead marks Bolshie Koty, the point of origin for the published transcriptome samples for the western species, while additional blue arrowhead marks Sukhoi Ruchei, the sampling point for the published *E. vittatus* and *E. cyaneus* mitochondrial genomes. The presumable geographic borders between the ranges are labeled. Insets: photographs of a typical representative of each species. All scale bars in the photographs are 10 mm. The S and W photos were taken with an Olympus Tough TG-5 digital camera under ceiling illumination, while the E photo was taken under an Altami SPM0880 microscope with side illumination and this seems slightly brighter. In all cases, color correction was performed using the same 17% gray paper. (**B**) COI haplotype networks of *E. verrucosus* individuals from the same locations based on the data published earlier [38]. A TCS network [63] is shown built with popart [64] v1.7 from the alignment of COI sequences of the corresponding samples [38]. Each hatch mark corresponds to one nucleotide substitution.

In this work, we further explore the cryptic species within *E. verrucosus*. We aimed to analyze mitochondrial genomes for a better insight in their phylogeny and genome sizes as a possible indicator of independent evolutionary history. Moreover, for the two cryptic species with adjacent ranges, we also experimentally tested the existence of a reproductive barrier. Our data show that all three cryptic species differ in their genome sizes, and at least the representatives of the western and southern species are separated with a post-zygotic reproductive barrier.

## 2. Results

### 2.1. Molecular Phylogeny and Difference in Genome Sizes Confirm Deep Genetic Separation between E. verrucosus Species

Previously, we found cryptic species within *E. verrucosus* based on the difference in sequences of COI and 18S gene fragments. In this work, we wanted to further explore the phylogenetic relationship within this group in comparison with two more congeneric species. One of them, *E. vittatus*, is one of the closest to *E. verrucosus* according to molecular phylogenetic studies [39,65]. In addition, *E. vittatus* also comprises at least two barcoding species [38]. The other, *E. cyaneus*, is relatively more distant [39,65] and was thus used as an outgroup. Both *E. vittatus* and *E. cyaneus* are sympatric with *E. verrucosus* in many of the locations. Thus far, there are no complete genomes for any of these species. However, there are available transcriptomes for all species from the same location (Bolshie Koty; W) [39,59,60,66] and also mitochondrial genomes for the western species of *E. verrucosus* from Bolshie Koty [61] and for *E. cyaneus* and *E. vittatus* from Sukhoi ruchei [67,68], which is separated from Bolshie Koty by the source of the Angara river (see the map in Figure 1). In this work, we sequenced transcriptomes for two *E. verrucosus* samples from Port Baikal (the southern species) and two samples from Ust-Barguzin (the eastern species). To be able to compare these data, we recovered the sequences of 15 mitochondrial genes (13 protein coding genes and two rRNA genes) for each species and location. Mitochondrial genes have been shown to be suitable for robust phylogenetic inference in gammaroid amphipods [67,69,70,71].

The obtained tree of 15 concatenated sequences placed the most recent common ancestor (MRCA) of the three *E. verrucosus* species at around 4.5 million years ago (Mya) and the MRCA of the W and S species at around 3.8 Mya (Figure 2A). Interestingly, the MRCA of *E. vittatus* samples, which originated from locations separated by Angara river source, was also estimated to be of comparable age. There have been suggestions that the rate of molecular evolution in Baikal amphipods is about five times higher that in other gammaroids based on the discrepancy between geological events and molecular evolution [37] or speciation rate [39]. However, even if we divide these values by this coefficient, we find that the divergence is much older than the appearance of the Angara river (50-60 thousands of years [72]).

The presence of barcoding species and the inconsistency between the cytology-based and sequencing-based estimates of genome size also led us to the decision to check genome sizes in the three lineages. According to our data, the genome size differed in all three pairwise comparisons with medians of 8.0 pg for the southern, 6.85 pg for the western, and 6.1 pg for the eastern species (Figure 2B). The value for the western lineage was consistent with the published flow cytometry-based value for this species [45] but not with the sequencing-based estimate [61]. Overall, these data provide additional support for the hypothesis of deep genetic separation of the species.

### 2.2. There Is a Reproductive Barrier between the Western and Southern Species

Presumably, the closest point of contact between the cryptic species of *E. verrucosus* is the source of the Angara river, where the western species (near the Listvyanka village) and the southern species (near Port Baikal) are separated by only about 1-km distance across the river. It was thus possible to sample animals in both locations and bring them to the laboratory at the same day.

The crossing experiment was performed the following way. We sampled 20 amplexuses at each point, separated males and females, and mixed the animals so that no animal could re-form amplexus with the same partner (Figure 3A). Amplexus formation started in all tanks almost immediately and lasted for about two months (Figure 3B). Then, ovigerous females started to appear in all tanks as well. They were not counted at each water exchange, as it is difficult to count ovigerous females exactly without too much handling due to the dark coloration of this species. However, we noticed that the number of females with eggs in their brood chambers in the mixed groups was gradually declining, and in two months from the start of the experiment we took the developing embryos from one female from each cross (i.e., each tank) for analysis and stained them with propidium iodide (PI) to visualize the nuclei.

We found that the stained embryos differed in their morphology. In the control crosses (W×W and S×S), the embryos were at different developmental stages and contained hundreds of visible nuclei at one side of the embryo; in many, the dorsal organ was clearly visible (Figure 3D and Appendix A). At the same time, in the experimental crosses W×S and S×W all embryos looked similar to each other and were most probably at the “soccerball” (S6) or “rosette” (S7) stages according to [74]; they contained from 34 to 57 visible nuclei at one side (Figure 3D and Appendix A), which is also consistent with these early stages of development [74].

At day 115 from the start of the experiment, juveniles emerged in the control groups (W×W and S×S), but they did not emerge in the mixed groups until the end of the experiment, day 182 (Figure 3C). Moreover, at the end of the experiment, all adult animals were closely analyzed, and no ovigerous females were found. Thus, we can conclude that the embryos stopped their development; most probably, they fell out of the brood pouch and were consumed by the adult animals.

## 3. Discussion

Cryptic species have received more and more attention during the last several decades [9,15,75]. They are important for solving the fundamental questions of genome evolution, as well as for such applied issues as biodiversity protection and biodiversity-related indices. In this work, we use the case of the Baikal species flock of endemic amphipods to provide new insights in the mechanism of speciation.

Here, we show that the cryptic species within *E. verrucosus* have probably split millions of years ago, differ in their genome size and at least two of them are separated by a reproductive barrier. As these species have non-overlapping ranges (at least now), we suppose that the differences in genome size are the consequence rather than the trigger of speciation. Thus far, it is unclear if these differences is purely accidental or were shaped by selection. The genome size of crustaceans is known to correlate on a multitude of factors, such as body size, temperature, habitat depth, developmental pattern, etc. Generally, the chain of consequences of greater genome size leading to larger cell size leading in turn to larger body size seems logical, but it is unclear if each of the other factors is a primary driver of genome size evolution or a confounding variable [76,77]. The body sizes of the three species have not been thoroughly compared, but in our experience, the usually sampled representatives of the eastern species are considerably smaller than the representatives of the other two species, which are similar to each other in size (see also Figure 1). The habitat depths are also generally similar (all species inhabit the littoral zone). The temperature regimes of the three habitat ranges seem to also be comparable, but we do not know when and where each species originated.

Mechanistically, the size of the genome can increase in two ways: either via whole genome duplication or via repeat expansion. In the case of *E. verrucosus*, the latter is most probably the case, as the differences are not even close to two-fold, and five *Eulimnogammarus* species or subspecies examined earlier had exactly the same karyotype, 2*n* = 52 [78]. Moreover, Baikal amphipods in general tend to have extremely stable karyotypes [78], while the genome size varies about 6-fold (from 2 to 12 pg) [45]. Indeed, the amount of repeats in the genome of *E. verrucosus* is very high, about 50% [61]. Thus, it is logical to assume that the genome size difference in *E. verrucosus* species is due to differential repeat expansion. It is interesting if there have been specific environmental factors promoting this difference.

This question leads us to the discussion of the time and possible triggers of speciation. Unfortunately, there are no fossil records of Baikal amphipods to calibrate the molecular clock. The COI difference within *E. verrucosus* is up to 13% [38]. The amphipod-specific substitution rate in COI is estimated as 0.01773 ± 0.004 substitutions per site per million years [73], which is close to the substitution rate of 2% established for primates [79] and commonly used as a universal value for animals. However, for deep-water Baikal amphipods, it was assumed to be elevated about five-fold [37,39]. According to the calibrated phylogeny (Figure 2A), the last common ancestor of the W and S species of *E. verrucosus* existed at the very least ≈600 thousand years ago (Tya), which is much earlier than the emergence of the Angara river, 60 Tya [72]. About 1-0.8 Mya, the level of the lake rose due to tectonic event and fell again after the formation of a new discharge through the Angara river [72]. Thus, between these events the littoral zone was larger and shifted in comparison to its current position. It is conceivable that the conditions in the past habitats of different populations of the ancestral *E. verrucosus* were more diverse and contributed to shaping their genomes.

The nature of the reproductive barrier is also an important question. In the case of western and southern *E. verrucosus*, the formation of precopulas did not seem to be inhibited in any way, as the number of precopulas was similar (at some points in time even higher) than in the control crosses (Figure 3B). We cannot rule out that some degree of mate discrimination may exist in the case an animal has a choice of conspecific and heterospecific mates, as our experimental design did not allow us to check for such possibility. However, our results confirm that the animals of both sexes perceive the representatives of the different cryptic species as their conspecifics.

There are several works, in which the authors assessed precopula formation, in some cases fertilization and embryo development between cryptic lineages or species in amphipods. In most cases, prezygotic isolation was found, contrary to our findings. In another gammaroid species, *G. fossarum*, in which the frequency of pairing was the higher, the more similar were the COI sequences in the male and female [20]. In a different study, it was found that the representatives of cryptic *Hyalella* species rarely formed interspecific precopulas; however, it is important to note that these species are sympatric [80]. In yet another different study, the males of cryptic species within *Paracalliope fluviatilis* were able to discriminate between “local” and “foreign” females and choose conspecific mate if they were collected >416 km apart and had genetic difference > 21.5% (in this case, the genetic and geographic distances were correlated) [81]. In contrast to these (and more similar to our work), close-to-random precopula formation was observed even within the co-occurring barcoding species of *Echinogammarus sicilianus* Karaman and Tibaldi, 1972 [31]; unfortunately, embryo development was not tracked in this work. The mode of incompatibility observed in *E. verrucosus*, i.e., embryos stalling at some point in their development, would clearly be detrimental to the fitness of the species due to the energy invested in reproductive effort. However, as these species presumably do not interact in the nature due to the geographic barrier, this loss of fitness does not occur and it is not selected against. In other words, pre-zygotic isolation is ensured by the geographic barrier. Taken together with the known geological record on changing lake levels in the last millions of years, these data corroborate the idea that the current border between the W and S species, the Angara river, is not the original cause for their separation.

Our data indicate that embryo development stops at the rosette stage. It corresponds to the time point where zygotic transcription becomes indispensable in the model amphipod *Parhyale hawaeinsis* [82]. This coincidence may mean that the hybrid embryos of *E. verrucosus* crosses stop their development because of some unbalanced factors in the zygotic genome. Unfortunately, there is not much information on embryo development in interlineage crosses in amphipods in the literature. In the study of *G. fossarum*, egg fertilization (assessed as embryos with >48 cells) was found to be a frequent event (>80%) if a precopula had formed [20]. This does not contradict our data, but as the authors did not monitor the further development of these embryos, it is hard to say if the mechanism observed in *E. verrucosus* also exists in *G. fossarum*.

## 4. Materials and Methods

### 4.1. Animal Sampling

Animals were collected with a hand net in: Listvyanka (51°52${}^{\prime }$14.07${}^{\prime \prime }$ N, 104°49${}^{\prime }$41.78${}^{\prime \prime }$ E; western species), Port Baikal (51°52${}^{\prime }$14.5${}^{\prime \prime }$ N 104°48${}^{\prime }$41.9${}^{\prime \prime }$ E; southern species), and Ust-Bargusin (53°22${}^{\prime }$29.56${}^{\prime \prime }$ N, 108°58${}^{\prime }$30.68${}^{\prime \prime }$ E; eastern species). The material used for RNA extraction or flow cytometry was shock frozen in liquid nitrogen either directly at the sampling spot or after transportation to the laboratory and kept at −80 °C until the analysis.

### 4.2. RNA Sequencing and Phylogenetic Analysis

Transcriptome sequencing was performed as described earlier [59] for two samples from Port Baikal (S) and two samples from Ust-Barguzin (E). The tissues of one whole animal were homogenized with Qiazol reagent Qiagen, Hilden, Germany) using a MM400 mixer mill (Retsch, Haan, Germany). Total RNA was isolated using the RNeasy Mini kit (Qiagen) and treated with DNase (TURBO DNA-free Kit, Thermo, Waltham, MA, USA). Then, mRNA was purified using the Oligotex mRNA Mini Kit (Qiagen). RNA concentration and quality were evaluated using a Qubit 2.0 fluorometer (Thermo) and a 2100 Bioanalyser instrument (Agilent, Santa Clara, CA, USA). Sequencing libraries were prepared with NEBnext Ultra II Directional Library Preparation Kit following the standard mRNA enrichment protocol with the Poly-A selection module (New England Biolabs, Hitchin, UK). Paired-end reads with a length of 100 bp were obtained with an Illumina HiSeq 2000 device. Twenty-four libraries (including those not analyzed in this study) were pooled and sequenced on four lanes of a v3 PE HiSeq flow cell. The sequencing data are available in NCBI (BioProject PRJNA871894). In addition to these four samples, three published *E. verrucosus* (W) and one *E. vittatus* transcriptome samples [39,60,66] were also taken into analysis.

The sequencing data were processed to extract mitochondrial genes. Our strategy combined the approaches used in [83,84]. First, the reads corresponding to the mitochondrial genome were extracted by aligning all reads to the reference mitochondrial genome of *E. verrucosus* KF690638 (W) [61] with bowtie2 [85] v2.3.5.1 using parameters suggested in MitoRNA [83]. Then, the selected reads were used to reconstruct mitochondrial genomes with MitoFinder [84] v1.4.1. Assemblies were performed with either metaSPAdes [86] v1.13.1 or MEGAHIT [87] v1.2.7. By default, metaSPAdes assemblies were used; in case a gene was incomplete or presumably erroneous nucleotides (not a multiple of three) were seen in the codon alignment (one case in poly(A) stretch in *ND3*), we checked it to the MEGAHIT assembly and alignment.

The sequences of 13 protein-coding mitochondrial genes and two ribosomal RNA genes were extracted and aligned with the same genes from complete mitochondrial genomes of three *Eulimnogammarus* species: *E. verrucosus* (W) KF690638 [61], *E. vittatus* KM287572 [68] and *E. cyaneus* KX341964 [67]. Sequences of each gene were aligned using mafft [88] v7.212 within UGENE [89] v33.0. The alignments were manually checked for assembly or annotation errors (see above) and trimmed to match the shortest sequence and also to the nearest codon in the case of protein-coding genes. The trimmed alignments were concatenated with catfasta2phyml [90], which also outputs a partition file. The best partitioning scheme and evolutionary model for each partition were identified with IQ-TREE [91,92] v2.3.1 using ModelFinder [93]; the set of models was restricted to JC, HKY, TN93, and GTR. This analysis returned three partitions. The first contained *atp6*, *atp8*, *cox1* (*COI*), *cox2*, *cox3*, *cytB*, *nd2* (*nad2*), *nd3* (*nad3*) and *nd6* (*nad6*), and the best model was GTR+F+G4. The second included *nd1* (*nad1*), *nd4* (*nad4*), *nd4L* (*nad4L*) with the same best model. The third consisted of the two rRNA genes, *rrnL* and *rrnS* with the best model being HKY+F+G4. These partitions were used to run analysis with BEAST [94] v2.6.6 with the site models mentioned earlier, except that *cox1* was used as a separate calibrated partition with a rate of 0.01773. The tree was fixed for all partitions, and the prior was set to the Yule model. The analysis ran for 10 million generations and yielded an effective sample size (ESS) value of 1593 for the desired TreeHeight parameter. The resulting trees were summarized with TreeAnnotator and visualized in FigTree [95] v1.4.4.

### 4.3. Flow Cytometry

Flow cytometry analysis was used for genome size estimation. Sample preparation was performed essentially according to [96]. Black soldier fly (*Hermetia illucens*) samples were used as a reference with known genome size. The estimated genome size of *H. illucens* is 1.16 Gb, or 1.18 pg, for both sexes [97,98]. Appendages from frozen *E. verrucosus* samples were mixed with appendages of *H. illucens* treated in the same way and homogenized in 100 µL of cold modified Galbraith buffer with RNase A [96] with a plastic pestle. Then, the volume was brought up to 1 mL with Galbraith buffer, and the homogenate was filtered through nylon mesh (≈20 × 20 µm holes) with a syringe. The samples were kept on ice whenever possible. Then, propidium iodide (PI, 1 mg/mL water solution) was added to the resulting nuclei suspension to the final concentration of 50 ppm, for staining overnight (about 16 h) at 4 °C in the dark. The next day, the fluorescence of the stained suspension was analyzed with CytoFlex S (Beckman Coulter, Brea, CA, USA) in the ECD-A channel. At least 3000 events (at least 10,000 events for most samples) were recorded, and populations were defined with the CytExpert software v2.4.0.28. The final genome size of each *E. verrucosus* sample was calculated as the ratio between median fluorescence of *E. verrucosus* and *H. illucens* diploid nuclei populations multiplied by the size of the *H. illucens* genome.

### 4.4. Crossing Experiment

For the crossing experiment, animals were collected in Listvyanka (the western species) and Port Baikal (the southern species) at the same day (11 October) at the water temperature of about 9 °C. This species reproduces once per year in autumn-winter [46,99], and most of the sampled animals were in the precopula stage. Precopulas were separated right after collection by gently detaching the male from the female with a plastic spoon. Then, the animals were mixed in the desired combinations, 10 males and 10 females per tank (Figure 3A), and transported to the laboratory. In the laboratory, the animals were kept at temperatures gradually (0.1 degree/day) falling to 6 °C and then at about 6 °C for the next six months. They were maintained in 2-L plastic tanks with approximately 1.5 L Lake Baikal water, three or four stones at the tank bottom, continuous aeration with water exchange and feeding ad libitum with a dried and ground mix of invertebrates and algae from the Baikal littoral twice a week. After the end of the experiment, the animals from the experimental aquaria were genotyped and sexed. For genotyping, we extracted genomic DNA from an appendage (two or three pleopods or two antennae) with the DNK-Sorb-M kit (Amplisens, Moscow, Russia) and performed PCR using two the pairs of primers with an HS-Taq reaction mix (Biolabmix, Novosibirsk, Russia). The first pair of primers, LWS4F (GCGGAACTGACTACCTCA) and LWS7R (CACGATGGGGTTGTAGAC) for an opsin gene, which anneal to conservative positions and produce a product in all *Eulimnogammarus* species tested [100]. The second pair of primers, Eve_F3 (AGAATAATCGGTACCTCTATAAGG) and Eve_R3 (GATTATGCCGAATGCAGGGAGGATG) [38], was found to anneal to the *cox1* gene in the animals from the western species but not from the southern species. The results of morphological sex determination and PCR-based genotyping completely coincided and indicated even distribution of males and females at the end of the experiment.

PI staining of the nuclei was performed the following way. First, embryos were removed from a female by water flow created with a Pasteur pipette. Then, they were fixed in acetone for 1–3 min and stained in the modified Galbraith buffer with 50 ppm PI (Sigma-Aldrich, Saint Louis, MO, USA, 81845) at +4 °C in the dark at least overnight.

Within the next two days, the nuclei distribution in the embryos was visualized with the CELENA S digital imaging system (Logos Biosystems, Anyang, South Korea) at the RFP channel (excitation 530/40; emission 605/55) under the 4× objective. The macro photographs of the embryos (Appendix A) were taken with an Olympus Tough TG-5 digital camera (Olympus, Hamburg, Germany). The number of nuclei was counted manually on contrast-corrected photographs with the GIMP software.

### 4.5. Data Analysis and Figure Preparation

Data analysis were performed in the R programming environment [101] v4.1.2 with openxlsx [102] v4.2.5 and other packages. The plots were also created with R using ggplot2 [103] v3.3.5, ggpubr [104] v0.4.0, and scales [105] v1.1.1; multipanel figures were assembled in Inkscape [106].

## 5. Conclusions

In this work, we found genome size differences for three (pseudo)cryptic species within the Baikal amphipod *E. verrucosus*. The genome size differences may be a consequence rather than a trigger of speciation, but they additionally confirm the genetic separation within the studied species.We also showed that at least the two species that are morphologically indistinguishable and have adjacent ranges are separated by a post-zygotic reproductive barrier. The presence of a post-zygotic barrier without an absolute pre-zygotic barrier is slightly unusual, as it would be detrimental to the fitness due to the energy invested in reproductive effort. However, it is explainable, as these species are separated with a geographic barrier, and thus this loss of fitness does not occur and is not selected against. The factors that determine the incompatibility are an interesting target for future work, as these could provide a new insight into the speciation mechanisms.Taken together, these data indicate that the previously applied barcoding approach indeed effectively indicated the separate biological species within *E. verrucosus*. These results provide new data for investigating genome evolution within relatively short times and also highlight the need for precise tracking of the sample origin in any studies in this morphospecies. In the case of the regions where the particular species is unknown (such as the Angara-Yenisei river system), it is highly desirable to determine the barcoding lineage for monitoring studies.

## Figures and Tables

**Figure 2 ijms-23-10858-f002:**
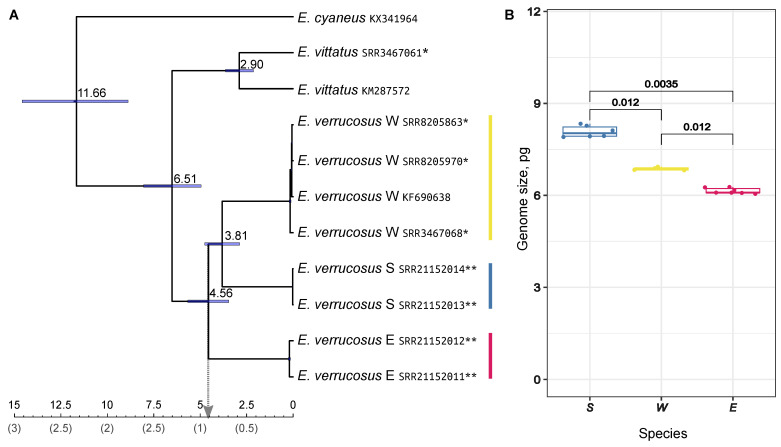
Phylogenetic relationship and genome sizes of *E. verrucosus* species. (**A**) A calibrated Bayesian phylogeny of three *Eulimnogammarus* morphospecies based on the sequences of 15 mitochondrial genes. Calibration (the upper scale) was based on the general substitution rate of 0.01773 in COI established for gammaroid amphipods [73]. The node bars show the 95% highest posterior density (HPD) intervals. The lower scale shows five-fold lower numbers based on the suggestion of elevated evolution rate in Baikal amphipods [37,39]. All posterior probability values were equal to 1 except for the *E. verrucosus* W SRR8205863 and SRR8205970. Single asterisks mark published transcriptome samples; double asterisks mark transcriptome samples obtained in this work; the samples without asterisks are published complete mitochondrial genomes. See Appendix A for references. (**B**) Genome size assessment for representatives of the three *E. verrucosus* species. The value in pg was estimated with flow cytometry as the ratio between median intensities of *E. verrucosus* and *Hermetia illucens* nuclei fluorescence. The numbers above the brackets represent *p*-values for the pairwise Wilcoxon rank sum test with Holm correction for multiple comparisons. The number of replicates: *n* = 4 for W; *n* = 6 for S and E. Please refer to Appendix A for raw data.

**Figure 3 ijms-23-10858-f003:**
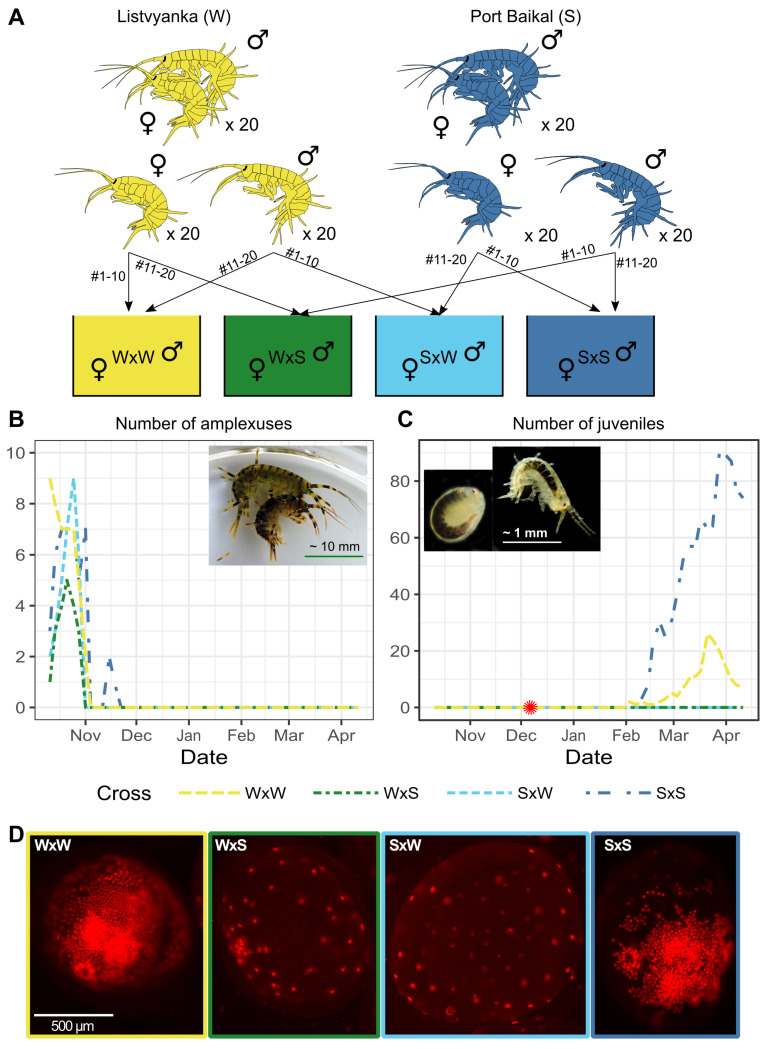
Experimental crossing. (**A**) The overview of the experimental design and designation of crosses; (**B**) the dynamics of amplexus formation. Inset: a typical amplexus; (**C**) the dynamics of juvenile hatching. Inset: an egg close to hatching and a newly hatched animal. The asterisk symbol marks the time point of embryo sampling. Please refer to Appendix A for raw counts; (**D**) a representative embryo from each cross, stained with a nucleic acid staining agent propidium iodide (PI) to visualize cell nuclei. For more photographs, see Appendix A.

## Data Availability

Raw sequencing data were submitted to NCBI (BioProject PRJNA871894). All other data obtained in this work is contained within the article and Appendix A. Custom scripts used for data analysis are available from https://github.com/drozdovapb/EveWxS_git/ (accessed on 23 August 2022).

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
