# Peer review of "Experimental Crossing Confirms Reproductive Isolation between Cryptic Species within Eulimnogammarus verrucosus (Crustacea: Amphipoda) from Lake Baikal"

_ijms, 2022, doi:10.3390/ijms231810858_

Round 1
Reviewer 1 Report
Major Concerns
I enjoyed reading this manuscript that highlights patterns of genome size and reproductive isolation in cryptic species of Eulimnogammarus verrucosus. While the work presented was interesting, I found it incomplete and narrow in scope. Measuring ecological variables that tend to be correlated with differences in genome size and a more complete assessment of reproductive isolation among pairwise comparisons of populations would greatly improve the manuscript. As is, the discussion is largely speculative.
For the crossing experiment, was survival 100%? If so, please state in the results. If not, please report and discuss how this affects interpretation of these results.
When discussing the development results, it would be more accurate to refer to individuals as embryos instead of eggs, I assume the pattern was that embyros in the marsupium were present for all crosses early in the experiment and that over time the inter-specific crosses had fewer and fewer embryos in the the marsupium. This is corroborated by embryos from the inter crosses having fewer nuclei = fewer cells and thus arrested development.
Minor Concerns
- For Figure 1 A, was the lighting for the images standardized. The eastern image seems much brighter.
- Please let the reader know the number of replicates that were used for Figure 1B. Hard to see the individual dots.
- There are several places where the writing is a bit awkward. Here are some examples: lines 90-92 and 131-133.
- Line 173- midgut anlage is not annotated on the supplement, figure.
- The examples in the text (lines 172-177) are not included in the figures. For example, there is no 900 nuclei embryo image and the inter crosses do not have a range to 57 nuclei (only 34-48 nuclei). I’m also curious why this figure is not included in the main text—also needs a figure caption.
- Figure 2D caption: Tell the reader what PI stands for and why it was used to stain the nuclei.
- Figure 2D is not cited in the results.
Author Response
Response to Reviewer 1 Comments
Point 1:
I enjoyed reading this manuscript that highlights patterns of genome size and reproductive isolation in cryptic species of Eulimnogammarus verrucosus. While the work presented was interesting, I found it incomplete and narrow in scope. Measuring ecological variables that tend to be correlated with differences in genome size and a more complete assessment of reproductive isolation among pairwise comparisons of populations would greatly improve the manuscript. As is, the discussion is largely speculative.
Response 1:
Thank you for your kind words!
We clearly agree that this work could be expanded but considered the results worth sharing. We totally agree that reproductive isolation should be tested in more detail, but as this species reproduces once per year, each change to the experimental design adds a year to the project. So far we do not even have direct evidence if the reproductive season in the eastern species coincides with that in the other two. However, we definitely plan to more completely assess the reproductive isolation further in this project. Thank you for this comment!
However, measuring ecological variables so far sounds (1) both unfeasible and (2) not necessarily applicable. (1) Among ecological factors, genome size is known to be correlated with temperature and habitat depths, and the latter seems to be very similar for all three species. A comprehensive comparison of their temperature regimes would require installation of numerous data loggers at multiple depths at least at several points of Baikal for at least a year. However, the ranges of latitudes are not that large and definitely intersect for the western and eastern popuations. (2) Now we expanded the manuscript with additional data in mitogenome- based phyologeny of these species (current Figure 2A; lines 153-180 in the Results section and 352-397 in the Methods section in the file with tracked changes), and the new data hints that the current geographical distribution of the species may actually be quite different from the locations, in which they emerged. We hope that future research will elucidate the evolutionary history of Baikal amhipods, bringing testable hypotheses on environmental effects on genome evolution.
Point 2:
For the crossing experiment, was survival 100%? If so, please state in the results. If not, please report and discuss how this affects interpretation of these results.
Response 2:
Unfortunately, the survival over the six-month experiment was not 100%. In three out of four aquaria, at least 17 out of 20 animals have survived. In one aquarium (females S x males W), unfortunately only 9 animals survived until the end of the experiment. However, we do not see how this could have affected the results. At the end of the experiment, we genotyped all the animals in experimental aquaria with PCR and determined the sex of each individual with morphological analysis. The distribution of animals by sex was even: 8 females : 9 males in the WxS cross and 4 females : 5 males in the SxW cross. We have added this information to the manuscript (lines 427-438 in the manuscript with tracked changes).
Point 3:
When discussing the development results, it would be more accurate to refer to individuals as embryos instead of eggs, I assume the pattern was that embyros in the marsupium were present for all crosses early in the experiment and that over time the inter-specific crosses had fewer and fewer embryos in the the marsupium. This is corroborated by embryos from the inter
crosses having fewer nuclei = fewer cells and thus arrested development.
Response 3:
Thank you for this comment! We have changed the text to refer to developing embryos wherever applicable.
Point 4:
Minor Concerns
1st For Figure 1 A, was the lighting for the images standardized. The eastern image seems much brighter.
Response 4:
Indeed, this particular photo was taken earlier and under brighter light conditions. We have added this information to the figure capture to avoid any confusion. Thank you for this suggestion! Unfortunately, it is impossible to take photos of animals of all three species at the same conditions at the moment. However, according to our observations, it is indeed true that the coloration of the animals from the eastern population is more yellowish.
Point 5:
2nd Please let the reader know the number of replicates that were used for Figure 1B. Hard to see the individual dots.
Response 5:
We have added the numbers in the figure caption (Figure 2B now) and are grateful for this suggestion.
Point 6:
3rd There are several places where the writing is a bit awkward. Here are some examples: lines 90-92 and 131-133.
Response 6:
Thank you for this comment! The sentence:
“Similarly, cryptic allozyme lineages correlated with geographic locations have been found for widely distributed shallow-water species E. cyaneus (Dybowsky, 1874) [33].”
now reads:
"Similarly, local populations of a widely distributed shallow-water species E. cyaneus (Dybowsky, 1874) were revealed to be cryptic allozyme lineages [33]."
This sentence:
“For this species, whole genome sequencing data have been collected, and based on these data, the genome size was estimated as 9.96 Gb based on selected single-copy gene coverage [60], or 10.18 pg if converted to mass units [61].”
now reads:
“For this species, whole genome sequencing was performed, and the genome size estimate based on the coverage of selected single-copy genes equaled 9.96 Gb [60] (10.18 pg if converted to mass units [61]).
We have also generally proofread the article and hope that it is now easier to read.
Point 7:
4th Line 173- midgut anlage is not annotated on the supplement, figure.
Response 7:
Thank you very much for this remark! We realized that it would be more correct to speak about the dorsal organ and the transverse rows of ectodermal cells, as these structures are much more clearly visible and also indicate a quite advanced post-gastrulation developmental stage (at least S11 Browne et al., 2005). We have marked these structures in Fig. S1 and added the description of the labels into the figure caption (lines 457-458 in the manuscript with tracked changes).
Point 8:
5th The examples in the text (lines 172-177) are not included in the figures. For example, there is no 900 nuclei embryo image and the inter crosses do not have a range to 57 nuclei (only 34-48 nuclei). I’m also curious why this figure is not included in the main text—also needs a figure caption.
Response 8:
We have decided to include a representative photograph of one embryo from each cross to the main figure and to provide a photo of each analyzed embryo as a supplementary material. We reasoned that the sample of embryos we were able to obtain was not enough for drawing statistical conclusions, but at the same time, showing all the photos was slightly confusing. Thus, the representative photos were chosen as somewhere in the range with most clearly visible arrangements of nuclei.
The captions for the supplementary information are provided after the main text of the manuscript (starting line 330 in the original document and line 456 in the revised document with tracked changes).
Point 9:
6th Figure 2D caption: Tell the reader what PI stands for and why it was used to stain the nuclei.
Response 9:
We have now added these details in the caption (Figure 3D now). Thank you for this comment!
Point 10:
7th Figure 2D is not cited in the results.
Response 10:
Thank you for this remark! We added the citation to the next.
Reviewer 2 Report
Baikalian gammarids are a well-known species flock established in one of the most ancient lake. Authors provide an excellent example of genomic incompatibility between cryptic species, based on results of experimental intra- and interspecies crossings. Additionally, DNA content was measured by flow cytometry and slight but significant differences were observed among “species”.
The presented result are counterintuitive to the common theory of speciation and has to be at least better discussed. The main point one can see from reading the paper that pre-mating isolation is absent and genetic barriers between two (West and South “species”) based entirely on post-zygotic isolation, most probably implemented as a form of incompatibility of the genomes. However, this is a no-choice experiment, which is not relevant to the real situation for this almost sympatric species (distance between two point were the species were collected is less than 1 km and separated by shallow Angara river). For the future research I would suggest to make two sets of experimental crossings implementing with-choice experiments,( i.e. 10 males of one species and 10+10 females of both "species"). Due to motphological similarity, it will be difficult to estimate number of conspecific and geterospecific amplexuses , but number of juveniles from females belonging to one or the other "species"might be easy scored by simple mtDNA PCR test. According to the most models for sympatric or parapatric speciation, we expect the development of not only post-zygotic, but also pre-zygotic isolation (mate recognition). Unfortunately, without with-choice experiments authors cannot rule out development of mate recognition among cryptic forms of E. verrucosus.
I would also suggest to bring forward to the main text the number of specimens for which the DNA content have been measured (this data can be found in S2 table) and how much variation were observed among specimens of one "species".
Author Response
Response to Reviewer 2 Comments
Point 1:
Baikalian gammarids are a well-known species flock established in one of the most ancient lake. Authors provide an excellent example of genomic incompatibility between cryptic species, based on results of experimental intra- and interspecies crossings. Additionally, DNA content was measured by flow cytometry and slight but significant differences were observed among “species”.
The presented result are counterintuitive to the common theory of speciation and has to be at least better discussed. The main point one can see from reading the paper that pre-mating isolation is absent and genetic barriers between two (West and South “species”) based entirely on post-zygotic isolation, most probably implemented as a form of incompatibility of the genomes. However, this is a no-choice experiment, which is not relevant to the real situation for this almost sympatric species (distance between two point were the species were collected is less than 1 km and separated by shallow Angara river). For the future research I would suggest to make two sets of experimental crossings implementing with-choice experiments,( i.e. 10 males of one species and 10+10 females of both "species"). Due to motphological similarity, it will be difficult to estimate number of conspecific and geterospecific amplexuses , but number of juveniles from females belonging to one or the other "species"might be easy scored by simple mtDNA PCR test. According to the most models for sympatric or parapatric speciation, we expect the development of not only post-zygotic, but also pre-zygotic isolation (mate recognition). Unfortunately, without with-choice experiments authors cannot rule out development of mate recognition among cryptic forms of E. verrucosus.
Response 1:
Thank you very much for this comment! We cannot totally agree that these species could be considered sympatric or almost sympatric. First, we have never sampled the representatives of the W and S species sympatrically yet. Second, while 1 km is indeed not a great distance, and the depth of the Angara river near its source is about 4-6 meters, but the velocity at the head of the river is 1-2 m/s or even higher if the water level is high [Galaziy, 1989, “Baikal in Questions and Answers” (in Russian), questions #894 and #895, retrieved from http://irkipedia.ru/content/angara_galaziy_g_i_baykal_v_voprosah_i_otvetah_1989]. According to our observations of the locomotor activity of E. verrucosus, their average locomotor activity, at least under laboratory conditions, is about 0.1 meters per second. Thus, most probably even if they attempt to cross the river, they get dragged by the current for a considerate distance. In this case each these species, indeed, might form a additional populations with unidirectional gene flow further along the river or even meet in reservoirs, which might be considered parapatric speciation if it actually takes place. However, it is important to note that this opportunity arose only about 50-60 thousands of years ago with the emergence of the Angara river, while according to molecular dating the S and W species split well before the emergence of Angara (even if we take into account the hypotheses on higher mutation rate in Baikal; see Fig. 2A and the corresponding text). Thus, these populations were separated before the Angara river became a barrier. Then, it is logical to assume that they might have formed in different locations from their current ranges, especially considering that before the emergence of Angara the level of Lake Baikal was considerably higher. This is why we do not find it counterintuitive that we did not see pre-zygotic reproductive isolation.
However, we vehemently agree that testing mate recognition would be expremely interesting. We have incorporated this information to the discussion and conclusions (lines 285-288 in the file with tracked changes). Thank you for the advice on the design! Unfortunately, this species only forms precopula once a year (we have now added this information into the Methods section, too, lines 418-419 in the file with tracked changes), so any change to the experimental
design is possible only once per year, making it unfeasible within this manuscript, but we will definitely run with-choice experiments next time.
Point 2:
I would also suggest to bring forward to the main text the number of specimens for which the DNA content have been measured (this data can be found in S2 table) and how much variation were observed among specimens of one "species".
Response 2:
We have added the numbers in the figure (now Figure 2B) caption and are grateful for this suggestion.
Reviewer 3 Report
This MS describe a study that distinguishes three cryptic species of Eulimnogammarus verrucosus from Lake Baikal. By using flow cytometry, this study confirmed that the E. verrucosus western, southern and eastern species (named by the collecting areas of Lake Baikal) have different genome size. Furthermore, a reproductive barrier was found between western and southern species. This study could provide some knowledge about genome evolution. Specific comments are as follows:
1. Because the genome sizes of three species are quite different (6.85pg, 8.0pg and 6.1pg), I am curious about the genetic differences in these three species. And the gene differences may be related to the resistance to different environmental stress. Is it possible to perform a genome analysis by next generation sequencing?
2. It is recommended to divide Materials and Methods into subsections and add subtitles to them.
3. This reviewer didn't see much of a problem with this MS. However, my feeling is that the current version of this MS doesn't seem to fit the main scopes of IJMS (i.e. biochemistry, molecular and cell biology, molecular biophysics, molecular medicine, and all aspects of molecular research in chemistry). If the authors can add a molecular biomarker study that can be used to distinguish between different species, or genomic analysis as mentioned above, it would become more suitable for this journal.
Author Response
Response to Reviewer 3 Comments
Point 1:
This MS describe a study that distinguishes three cryptic species of Eulimnogammarus verrucosus from Lake Baikal. By using flow cytometry, this study confirmed that the E. verrucosus western, southern and eastern species (named by the collecting areas of Lake Baikal) have different genome size. Furthermore, a reproductive barrier was found between western and southern species. This study could provide some knowledge about genome evolution. Specific comments are as follows:
- Because the genome sizes of three species are quite different (6.85pg, 8.0pg and 6.1pg), I am curious about the genetic differences in these three species. And the gene differences may be related to the resistance to different environmental stress. Is it possible to perform a genome analysis by next generation sequencing?
Response 1:
Unfortunately, running detailed whole genome analysis is quite tricky due to the comparatively large genome sizes and large proportion of repeats. So far, the attempts to do it are definitely among our main goals, but it does not seem feasible in the framework of this manuscript. However, we were able to add mitochondrial genome analysis to the manuscript (current Figure 2A; lines 153-180 and 352-397 in the file with tracked changes) to provide a further insight into the phylogeny and evolution of these species. Thank you for this suggestion!
Point 2:
- It is recommended to divide Materials and Methods into subsections and add subtitles to
Response 2:
We have done so now. Thank you for this suggestion!
Point 3:
- This reviewer didn't see much of a problem with this MS. However, my feeling is that the current version of this MS doesn't seem to fit the main scopes of IJMS (i.e. biochemistry, molecular and cell biology, molecular biophysics, molecular medicine, and all aspects of molecular research in chemistry). If the authors can add a molecular biomarker study that can be used to distinguish between different species, or genomic analysis as mentioned above, it would become more suitable for this
Response 3:
We have now added a section on the analysis of mitochondrial genomes (current Figure 2A; lines 153-180 in the Results section and 352-397 in the Methods section in the file with tracked changes) of the studied species with new sequencing data obtained in this project and not published earlier. We are very grateful for this suggestion and hope that it significantly improved the manuscript in general, as well as its fit to the main scopes of IJMS. Our data go in line and continue the ongoing conversation about the evolution of mitochondrial genomes of amphipods published in the journal recently (Mamos et al., 2021 https://doi.org/10.3390/ijms221910300).
Round 2
Reviewer 1 Report
The authors have adequately addressed my concerns.
Reviewer 3 Report
The author has answered most of my questions well. I think the current version of this MS can be accepted by IJMS.